

# Parameter estimation in tree graph metabolic networks

Laura Astola[1], Hans Stigter[2], Maria Victoria Gomez Roldan[3], Fred van Eeuwijk[2], Robert D. Hall[4], Marian Groenenboom[2] and Jaap J. Molenaar[2]

[1] Department of Biomedical Engineering, Eindhoven University of Technology, Eindhoven, Netherlands
[2] Biometris, Department for Mathematical and Statistical Methods, Wageningen University and Research Centre, Wageningen, Netherlands
[3] Institute of Plant Sciences Paris-Saclay, Gif-sur-Yvette, France
[4] Plant Research Intenational—Bioscience, Wageningen University and Research Centre, Wageningen, Netherlands

Corresponding author
Laura Astola, l.j.astola@gmail.com, l.j.astola@tue.nl

## ABSTRACT

We study the glycosylation processes that convert initially toxic substrates to nutritionally valuable metabolites in the flavonoid biosynthesis pathway of tomato (*Solanum lycopersicum*) seedlings. To estimate the reaction rates we use ordinary differential equations (ODEs) to model the enzyme kinetics. A popular choice is to use a system of linear ODEs with constant kinetic rates or to use Michaelis–Menten kinetics. In reality, the catalytic rates, which are affected among other factors by kinetic constants and enzyme concentrations, are changing in time and with the approaches just mentioned, this phenomenon cannot be described. Another problem is that, in general these kinetic coefficients are not always identifiable. A third problem is that, it is not precisely known which enzymes are catalyzing the observed glycosylation processes. With several hundred potential gene candidates, experimental validation using purified target proteins is expensive and time consuming. We aim at reducing this task via mathematical modeling to allow for the pre-selection of most potential gene candidates. In this article we discuss a fast and relatively simple approach to estimate time varying kinetic rates, with three favorable properties: firstly, it allows for identifiable estimation of time dependent parameters in networks with a tree-like structure. Secondly, it is relatively fast compared to usually applied methods that estimate the model derivatives together with the network parameters. Thirdly, by combining the metabolite concentration data with a corresponding microarray data, it can help in detecting the genes related to the enzymatic processes. By comparing the estimated time dynamics of the catalytic rates with time series gene expression data we may assess potential candidate genes behind enzymatic reactions. As an example, we show how to apply this method to select prominent glycosyltransferase genes in tomato seedlings.

## INTRODUCTION

In this paper we study metabolic network inference from given biological time-series data. The two main ingredients in general metabolic pathway inference are the reconstruction of the network topology and the estimation of the parameters involved. When the network is large and the concentrations of intermediates are unknown, or when there are no time series data available, one may still study the fluxes by setting up stoichiometric models for flux balance analysis (*Varma & Palsson, 1995*; *Stelling et al., 2002*; *Orth, Thiele & Palsson, 2010*). If time-series data of metabolites are available ordinary differential equations (ODEs) can often provide a suitable model (*Chen, Niepel & Sorger, 2010*; *Chou & Voit, 2009*; *Srinath & Gunawan, 2010*; *Hatzimanikatis, Floudas & Bailey, 1996*). If the enzymes involved are also known, it is customary to use enzyme-kinetic models (*Steuer & Junker, 2009*; *Schallau & Junker, 2010*; *Liebermeister & Klipp, 2006*) with Michaelis–Menten kinetics, although the reliability of this approach has been questioned, especially when applied to *in vivo* measurements (*Savageau, 1995*; *Hill, Waight & Bardsley, 1977*). When (part of) the catalytic rates are not known, linear ODEs (*Astola et al., 2011*) and general biochemical systems theory (*Voit, Marino & Lall, 2005*) can be used. When the network topology is completely unknown, the situation is more complicated, although some recent studies attempt to tackle this problem using methods based on genetic algorithms (*Schmidt et al., 2011*). Still, the uniqueness of the reconstructed network is often compromised and the identifiability of the system remains an issue that needs to be investigated (*Craciun & Pantea, 2008*; *Srinath & Gunawan, 2010*). Model identifiability is an essential prerequisite in making any conclusions from (by default limited number of) observations. The foremost categories of identifiability are the structural and the practical identifiability, the former related to the symbolic expression of the model itself and the latter related to the amount and nature of the available data. We will test our models and data on both conditions.

Here we discuss a special and relevant class of network topologies, which are so-called tree networks and show that in such networks linear models yield parameter estimates that are unique in the structural sense. As the name suggests, a tree graph looks like a branching tree where the edges (arrows) are directed so that the nutrients flow from root to leaf (cf. Fig. 1).

As in real trees the branches do not form cycles. By a cycle we mean any closed chain of edges regardless of the directions of the edges. In many biological pathways, such as in the flavone and the flavonol biosynthesis (*Kanehisa Laboratories, 2010*), a tree graph captures the network of the enzymatic reactions. Indeed metabolic networks with tree structures constitute a relevant class including, for example, large parts of the biosynthetic pathways of, e.g., $\gamma$-carotene, limonene, ansamycin and puromycin etc. (*Kanehisa Laboratories, 2010*).

Although this paper focuses on the mathematical modeling of tree structured metabolic networks in general, the original motivation rose from biological questions concerning the specific networks in flavonol biosynthesis. Therefore we have also included a brief Material and Methods section to refer to the original data generated prior to this study. The paper

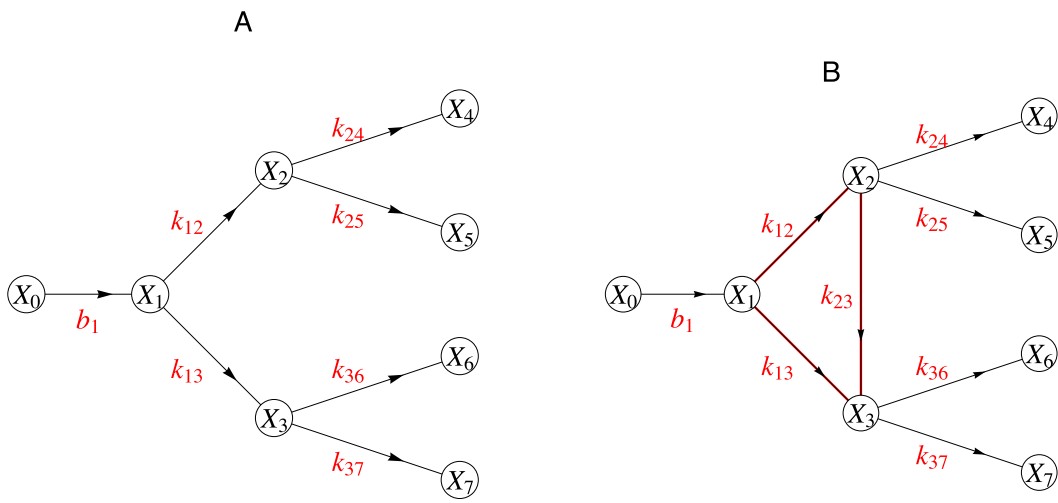

**Figure 1** (A) A graph with a tree structure. (B) This graph contains a cycle and is thus not a tree graph. The catalytic rate corresponding to reaction between node $i$ and $j$ is indicated as $k_{ij}$. Here the node $X_0$ represents a boundary node connecting this network to the surrounding larger network.

is organized as follows: in 'Parameter estimation in general networks' to set the stage we review our earlier work in modeling metabolic pathways using time-invariant systems of linear differential equations and discuss the particular properties of tree-graph networks. In 'Discussion' we consider the essential problem of model identifiability and show that our candidate networks satisfy the criteria for structural and practical identifiability. In 'Time varying kinetic rates' we propose a novel application for our time-variant estimation scheme by showing how it can be employed in finding the most likely catalysts from a large set of enzymes.

## MATERIALS AND METHODS

Throughout this article we use as a model example data the time series of the concentrations of the metabolites involved in a putative quercetin glycosylation pathway (*PlantCyc, 2016*). The data explored and modelled in this article originates from the research by *Gomez Roldan et al. (2014)*, where flavonol pathway related metabolites were studied in tomato seedlings. The metabolites were measured from roots, hypocotyls, and cotyledons on different days and under different conditions. The time series of metabolite concentration data that we used in the mathematical models were statistically corrected for fixed and random effects with a standard mixed model pre-processing resulting in the so-called best linear unbiased predictions (BLUP) and provided as Supplementary Data. (In SAS this can be done with the command: Proc Mixed.) The original metabolite concentration time series and the corresponding enzymatic assays are included in the Supplementary Data. The supplementary data also contains Mathematica notebooks to estimate the kinetic rates from data and to do sensitivity analysis of the reconstructed model. In this section we further discuss the theoretical analysis and how we implement the practical parameter estimation on metabolic networks.

## Parameter estimation in general networks

We first consider the parameter estimation problem in general linear time-invariant ordinary differential equation (LTI-ODE) systems. For convenience, we briefly sketch the approach when the catalytic rates are constants over time as in our previous work (*Astola et al., 2011*).

We recall that any network can be represented as a graph, where nodes are connected by edges when there is some interaction between these nodes. In a metabolic network a node represents a substrate or a product, and a directed edge from node $i$ to node $j$ means that $i$ can be converted to $j$ by enzymatic activity. To an edge from node $i$ to $j$, we assign a weight, i.e., the catalytic rate $k_{ij} \geq 0$, which represents the rate of product formation. In parameter inference one estimates the $k_{ij}$ from data.

Denoting the concentration of substrate $i$ at time $t$ as $X_i(t)$, a general time-invariant linear ODE model with a constant nonhomogeneous term, satisfying the mass conservation law, can be written as

$$\dot{X}_i(t) = -\sum_{j \neq i} k_{ij} X_i(t) + \sum_{j \neq i} k_{ji} X_j(t) + b_i, \tag{1}$$

for $i = 1, \ldots, n$, with

$$b_i = \begin{cases} \text{constant} & \text{if } i = 1 \\ 0 & \text{otherwise.} \end{cases} \tag{2}$$

The first summation in (1) stands for the edges leaving $X_i$, the second for the incoming edges and $b_i$ for the possible in or outflow to the system. To simplify the notation, we introduce a matrix $A$ with components given by

$$\begin{cases} A_{ij} = k_{ji}, & i \neq j \\ A_{ii} = -\sum_{j \neq i} k_{ij}. \end{cases} \tag{3}$$

Then, (1) becomes

$$\dot{X}_i(t) = \sum_{j=1}^{n} A_{ij} X_j(t) + b_i, \quad i = 1, \ldots, n. \tag{4}$$

Equation (4) can be rewritten in a compact matrix form as

$$\dot{X}(t) = \begin{pmatrix} \dot{X}_1(t) \\ \dot{X}_2(t) \\ \vdots \\ \dot{X}_n(t) \end{pmatrix} = \begin{pmatrix} -\sum_{j \neq 1} k_{1j} & k_{n1} & b_1 \\ k_{12} & k_{n2} & 0 \\ \vdots & & \vdots \\ k_{1n} & -\sum_{j \neq n} k_{nj} & 0 \end{pmatrix} \begin{pmatrix} X_1(t) \\ X_2(t) \\ \vdots \\ X_n(t) \\ 1 \end{pmatrix} = \tilde{A} \cdot \tilde{X}(t), \tag{5}$$

where $\tilde{X}(t)$ is obtained from $X(t)$ by appending an extra 1 and matrix $\tilde{A}$ is obtained from $A$ by extending it with an extra column containing the constant $b_1$.

To reconstruct a metabolic network from time-series measurements, we have to estimate the reaction rates $k_{ij}$, i.e., the weights of the edges in the network and the flow terms $b_i$. In view of (5), it is sufficient to estimate the $(n+1)\times(n+1)$ matrix $\tilde{A}$. We denote the data, i.e., measured concentrations of substrate $i$ at time points $t_j$, $j = 1,\ldots,m$, as an $(n\times m)$ matrix $\mathbb{X}$. Estimates of the derivatives of the data curves we will store in a matrix $\dot{\mathbb{X}}$. To compute these estimates we may proceed in two ways. First, construct two $n\times m$ data matrices $\mathbb{X}_0, \mathbb{X}_1$ as follows

$$\mathbb{X}_0 = \begin{pmatrix} \mathbb{X}_{1,m-1} & \mathbb{X}_{1,m-2} & \ldots & \mathbb{X}_{1,0} \\ \mathbb{X}_{2,m-1} & \mathbb{X}_{2,m-2} & \ldots & \mathbb{X}_{2,0} \\ \vdots & & \ldots & \vdots \\ \mathbb{X}_{n,m-1} & \mathbb{X}_{n,m-2} & \ldots & \mathbb{X}_{n,0} \end{pmatrix}, \ \mathbb{X}_1 = \begin{pmatrix} \mathbb{X}_{1,m} & \mathbb{X}_{1,m-1} & \ldots & \mathbb{X}_{1,1} \\ \mathbb{X}_{2,m} & \mathbb{X}_{2,m-1} & \ldots & \mathbb{X}_{2,1} \\ \vdots & & \ldots & \vdots \\ \mathbb{X}_{n,m} & \mathbb{X}_{n,m-1} & \ldots & \mathbb{X}_{n,1} \end{pmatrix}, \tag{6}$$

where $m$ is the number of measurements. The matrix

$$\dot{\mathbb{X}} \equiv \frac{1}{\Delta t}(\mathbb{X}_1 - \mathbb{X}_0), \tag{7}$$

could then be used as an approximation for $\dot{X}$. For simplicity we assume the time grid to be equidistant with time step $\Delta t$. If this is not the case, the necessary modifications are easily implemented.

Secondly, we may use an alternative and often better approach to obtain approximations for $\dot{\mathbb{X}}_i$ by fitting splines to the time series data $\mathbb{X}_i$ (*Zhan & Yeung, 2011*). To obtain curves that interpolate the data faithfully, we require that the distances between the curves and the measurements are minimal and that at the same time the curves are smooth. To achieve this we fit P-splines, which are B-splines with a penalization for non-smoothness (*Eilers & Marx, 1996*). From these splines, we evaluate the derivative estimates at time points $t_j$. These estimates are then used as entries in the matrix $\dot{\mathbb{X}}$. Having at hand an estimate for matrix $\dot{\mathbb{X}}$, the problem of network inference comes down to finding the matrix $\tilde{A}$ from the equation

$$\dot{\mathbb{X}} = \tilde{A}\tilde{\mathbb{X}}, \tag{8}$$

in which $\dot{\mathbb{X}}$ is known and $\tilde{\mathbb{X}}$ is obtained from the data matrix $\mathbb{X}$ by extending this with an extra row of ones. However, solving $\tilde{A}$ directly from (8) often results in over-fitting, since all possible edges are included in the modeled network. Another serious shortcoming of such a matrix (pseudo-) inversion approach is the fact that we cannot control the positivity of the reaction rates. Although in *Schmidt, Cho & Jacobsen (2005)* negative coefficients were interpreted as inhibition of the compounds, in many biological pathways, negative coefficients are not permitted. Thus we take a more general approach in which one can exclude all edges that are biologically not acceptable, and in which one can constrain the reaction rates to be positive, without substantially compromising computation time.

To this end, we reformulate the equation as a minimization problem:

$$\underset{\tilde{A}}{\mathrm{argmin}}\left(\| \dot{\mathbb{X}} - \tilde{A}\tilde{\mathbb{X}} \|\right). \tag{9}$$

The matrix norm used here is the Frobenius norm:

$$\|\tilde{A}\| = \sqrt{\sum_{i=1}^{n}\sum_{j=1}^{m}\tilde{A}_{ij}^2}. \tag{10}$$

This alternative formulation allows inclusion of expert knowledge in a simple way. We put $\tilde{A}_{ij} = 0$, when an edge from node $i$ to node $j$ cannot exist. Nearly all mathematical software packages (Mathematica, Matlab, Maple, etc.) can numerically find the minimizer $\tilde{A}$ (and thus the reaction rates $k_{ij}$ and the flow term $b_1$) with the constraint that $k_{ij} \geq 0$.

## Parameter estimation in tree networks

As described in the introduction, tree networks are networks, whose graphs resemble trees in that they branch away from the root and the directions of the edges always point from the root towards the leaves. In Fig. 1 we presented, using an example, the difference between a tree and a non-tree graph. In a kinetic reaction system with a tree network, the parameters can be uniquely estimated even when they are time dependent. We could write this down in general. However, the proof is based on one central idea. We feel that the reader gains more insight if we simply show this idea through an example. To that end we use as example the network in the left hand side of Fig. 1. The extension to the general is straightforward.

For the network on Fig. 1A, we have the following kinetic mass balance model:

$$\begin{pmatrix}\dot{X}_1 \\ \dot{X}_2 \\ \dot{X}_3 \\ \dot{X}_4 \\ \dot{X}_5 \\ \dot{X}_6 \\ \dot{X}_7\end{pmatrix} = \begin{pmatrix} -(k_{1,2}+k_{1,3}) & 0 & 0 & 0 & 0 & 0 & 0 & 0 & b_1 \\ k_{1,2} & -(k_{2,4}+k_{2,5}) & 0 & 0 & 0 & 0 & 0 & 0 & 0 \\ k_{1,3} & 0 & -(k_{3,6}+k_{3,7}) & 0 & 0 & 0 & 0 & 0 & 0 \\ 0 & k_{2,4} & 0 & 0 & 0 & 0 & 0 & 0 & 0 \\ 0 & k_{2,5} & 0 & 0 & 0 & 0 & 0 & 0 & 0 \\ 0 & 0 & k_{3,6} & 0 & 0 & 0 & 0 & 0 & 0 \\ 0 & 0 & k_{3,7} & 0 & 0 & 0 & 0 & 0 & 0 \end{pmatrix}\begin{pmatrix}X_1 \\ X_2 \\ X_3 \\ X_4 \\ X_5 \\ X_6 \\ X_7 \\ 1\end{pmatrix}, \tag{11}$$

where the constant $b_1$ represents the influx into the system and the $k_{i,j}$ are the catalytic rates. Note that there are as many unknown parameters $(k_{i,j}, b_1)$ as there are measured variables $X_i(t_j)$. Therefore, as can be directly verified, we can rewrite the previous matrix equation by exchanging the $X_i$ and $k_{i,j}$ as follows:

$$\begin{pmatrix}\dot{X}_1 \\ \dot{X}_2 \\ \dot{X}_3 \\ \dot{X}_4 \\ \dot{X}_5 \\ \dot{X}_6 \\ \dot{X}_7\end{pmatrix} = \underbrace{\begin{pmatrix} 1 & -X_1 & -X_1 & 0 & 0 & 0 & 0 \\ 0 & X_1 & 0 & -X_2 & -X_2 & 0 & 0 \\ 0 & 0 & X_1 & 0 & 0 & -X_3 & -X_3 \\ 0 & 0 & 0 & X_2 & 0 & 0 & 0 \\ 0 & 0 & 0 & 0 & X_2 & 0 & 0 \\ 0 & 0 & 0 & 0 & 0 & X_3 & 0 \\ 0 & 0 & 0 & 0 & 0 & 0 & X_3 \end{pmatrix}}_{\text{matrix } B}\begin{pmatrix}b_1 \\ k_{1,2} \\ k_{1,3} \\ k_{2,4} \\ k_{2,5} \\ k_{3,6} \\ k_{3,7}\end{pmatrix}. \tag{12}$$

We immediately see that $B$ is an upper triangular matrix since the entries below the diagonal are zero. This implies that the determinant of the matrix $B$ in (12) is the product

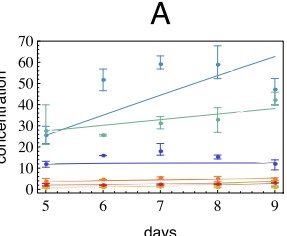 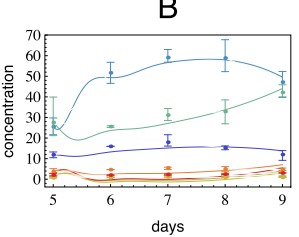 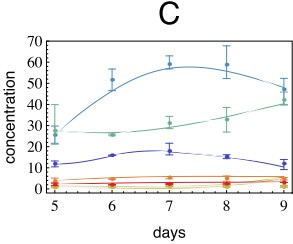

**Figure 2** **In this figure we have used three different models to reconstruct a flavonol concentration data indicated as dots.** The compounds shown here belong to a pathway with putative structure as on the (A) in Fig. 1A. The colors of the reconstructed curves correspond to those of the dots. (A) A reconstruction with a tree network and constant catalytic rates. (B) A reconstruction with the full network (all nodes are connected to each other) and constant catalytic rates. Note that the fit is still poor, although the number of parameters is much higher than in the case on the left. (C) A reconstruction with the same tree structure as in (A), but with time dependent catalytic rates.

of the entries on the diagonal: $X_1^2 \cdot X_2^2 \cdot X_3^2$, and thus unequal to 0 since $X_i \neq 0, \forall i = 1, \dots, n$. So, $B$ is invertible and the system of equations has the unique solution.

$$
\begin{pmatrix} b_1 \\ k_{1,2} \\ k_{1,3} \\ k_{2,4} \\ k_{2,5} \\ k_{3,6} \\ k_{3,7} \end{pmatrix} = \underbrace{\begin{pmatrix} X_0^{-1} & X_0^{-1} & X_0^{-1} & X_0^{-1} & X_0^{-1} & X_0^{-1} & X_0^{-1} \\ 0 & X_1^{-1} & 0 & X_1^{-1} & X_1^{-1} & 0 & 0 \\ 0 & 0 & X_1^{-1} & 0 & 0 & X_1^{-1} & X_1^{-1} \\ 0 & 0 & 0 & X_2^{-1} & 0 & 0 & 0 \\ 0 & 0 & 0 & 0 & X_2^{-1} & 0 & 0 \\ 0 & 0 & 0 & 0 & 0 & X_3^{-1} & 0 \\ 0 & 0 & 0 & 0 & 0 & 0 & X_3^{-1} \end{pmatrix}}_{\text{matrix } B^{-1}} \begin{pmatrix} \dot{X}_1 \\ \dot{X}_2 \\ \dot{X}_3 \\ \dot{X}_4 \\ \dot{X}_5 \\ \dot{X}_6 \\ \dot{X}_7 \end{pmatrix}. \tag{13}
$$

## Time varying kinetic rates

In earlier work we developed a fast method to reconstruct metabolic networks (*Astola et al., 2011*). The idea in this approach was to substitute the measurements directly into the model equations and not only in the objective function. This approach had as a limitation that all parameters were assumed to be constant in time. Here we extend our previous approach by allowing the catalytic rates to be time dependent, to better reflect the real situation, since in practice the enzyme concentrations are fluctuating in time. This has also immediately resulted in reconstructions that better fit the observed data as can be seen in Fig. 2. While the standard practice in enzyme kinetics is to either use constant catalytic rates in mass balance equation or to model product formation through a Hill function (*Goutelle et al., 2008*) such as in the Michaelis–Menten equation (*Savageau, 1995*), none of these take into account the fact that the enzyme concentration is also changing in time. Since we also want to study the relation of gene expression and enzyme concentration in time, we need to capture their dynamics.

As the catalytic rate is now modeled as a function in time, and not as a constant, it is no longer possible to infer this with the standard procedure of solving for those parameters that fit the ordinary differential equations to data in the sense of maximum likelihood. We cannot clearly separate the substrate/product, enzyme concentrations and noise, since

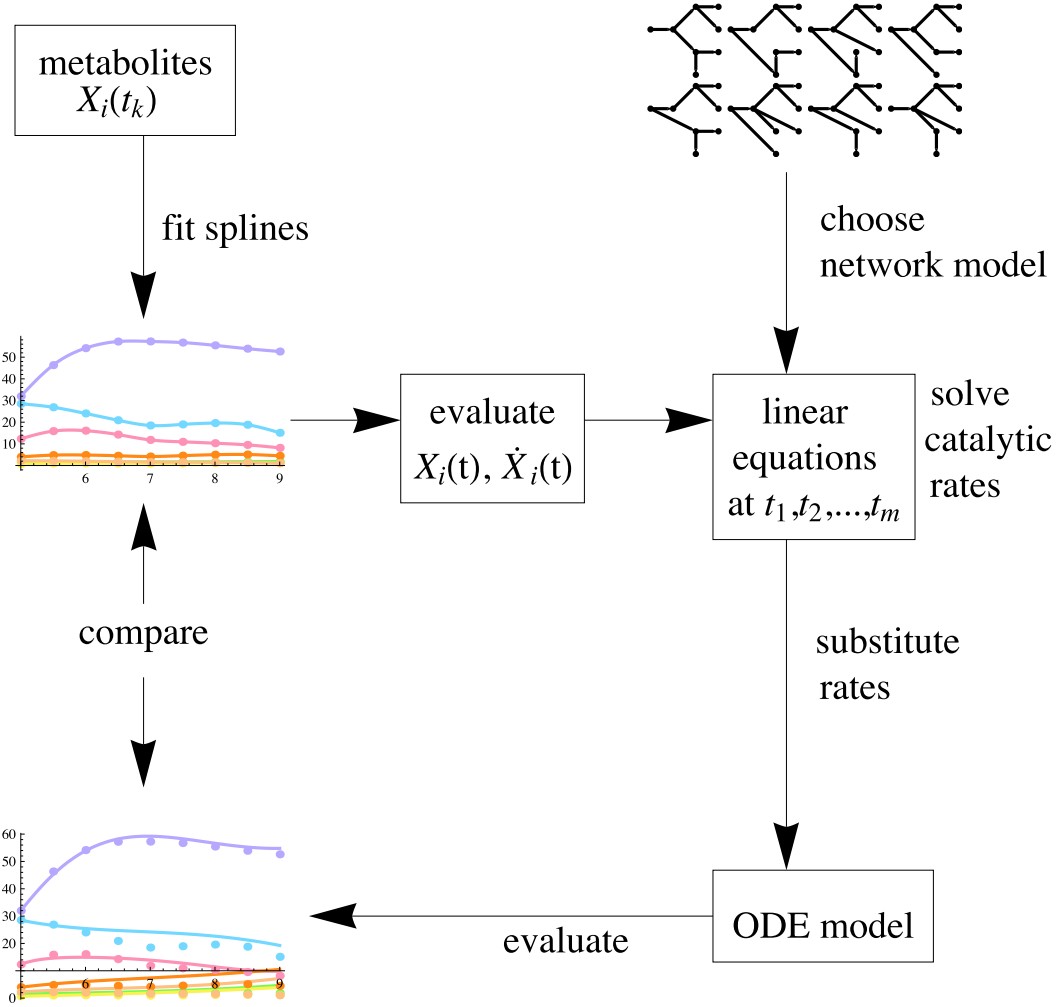

**Figure 3  A schematic view of the inference procedure.** After fitting splines to data, the parameters can be estimated for any given network of choice. Next, the optimal network can be selected by comparing the reconstruction result with each candidate network model to the original measurements.

we have no measurements of the enzyme concentrations. To solve them, we would have to impose a model on them, which we don't have a priori. A reasonable approach in this situation is to first estimate a model for the metabolite concentrations for which we have several measurements. By fixing the concentrations first using spline approximations, we may then estimate the trends in the enzyme concentrations. This method assumes that the solutions are rather smooth. If this is not the case and the sampling frequency is low, the derivatives obtained by fitting splines can introduce errors that distort the reconstruction. The inference method proposed here is by no means restricted to tree networks, but in case the network has a tree structure, the parameters can be estimated in an unambiguous way. We summarize the general work flow for the proposed parameter inference in the schematic diagram in Fig. 3.

## Time dependent parameter estimation

In this section we present three different schemes to estimate the $k_{ij}(t)$ in model (4). In (9) we used the data at all time points simultaneously to estimate the time independent parameters. However, a remarkable feature of tree structured networks is that the data at one time point is already enough to calculate unique estimates for the parameter values at that particular time point. This is immediately clear from (13): as soon as we have estimates for the time derivatives $\dot{X}(t_k)$ available, we may calculate estimates for the $k_{ij}(t_k)$.

Scheme 1. To estimate the derivatives at some time point one still needs the data of neighboring time points. So, the first step in this scheme is to fit, e.g., P-splines to the data time series (*O'Sullivan, 1986*; *Eilers & Marx, 1996*). From these splines we calculate estimates for the time derivatives $\dot{X}_i(t_k)$. Then by substituting these estimates as well as the measurements into Eq. (4), we are left with a set of linear equations to solve $k_{ij}(t_k)$ and $b_1$ at all times $t_k$. Finally, for smooth and continuous catalytic rates, one may fit, e.g., a second order polynomial through these estimates.

Scheme 2. An alternative approach in which the number of parameters is smaller than in scheme 1, is to assume that the functions $k_{ij}(t)$ can be adequately represented as polynomials in time of some order. In practice order 2 is often sufficient. With this choice we have then:

$$k_{ij}(t) = \alpha_{ij} t^2 + \beta_{ij} t + \gamma_{ij}. \tag{14}$$

This implies that per $k_{ij}$ we have 3 parameters to be estimated using the whole time series data. By substituting (14) into matrix $\tilde{A}$ in (9) we then obtain estimates for $\alpha_{ij}$, $\beta_{ij}$ and $\gamma_{ij}$, and thus for $k_{ij}(t)$.

Scheme 3. As in the previous scheme, we assume (14). We construct an objective function like the following:

$$\sum_k \| X(t_k) - \mathbb{X}(t_k) \|, \tag{15}$$

which is the sum of the distances between $X(t_k)$ and the measurements. We look for a matrix $\tilde{A}$, such that the solutions $X_i(t)$ to (4) minimize this objective function. Using suitable optimization algorithm we simultaneously estimate $X_i$, $k_{ij}$, and $b_1$.

To compare the fit, accuracy and speed of these three schemes we applied them using as test networks random tree networks that have equal numbers of nodes and edges as the network on Fig. 1A.

In these networks, we simulated time series data with time varying catalytic rates. To generate artificial data, we assigned random values to $\alpha_{ij}, \beta_{ij}$ and $\gamma_{ij}$ in a range, such that the resulting solutions have approximately the same range as the metabolite concentration data for quercetin glycosides measured in tomato seedlings (cf. Fig. 2). To assess the reconstruction power of the three schemes, we also tested them on networks that are not trees. The corresponding data generation process is the same but the network models contain cycles. In the third set of simulations we added $\pm 10\%$ uniformly distributed noise to tree structured network data.

## Parameter inference as a mean to select active genes

In addition, as a potentially powerful application, we show how we may infer the gene candidates likely to be involved in the enzymatic reactions. This can be done by comparing estimated time dependent catalytic rates with simultaneously measured gene expression data. If, according to the model, the formation of a metabolite necessitates higher/lower enzyme concentration, this should be also observable in the expression level of the gene that codes for this enzyme. Using this heuristic, we were able to select from a large set of potential genes the most likely candidate genes for further experimental validation of their functioning in particular reactions. In view of this application, small inaccuracies in parameters are not detrimental, since here we are mainly interested the dynamic trends of the catalytic rates instead of their precise numeric values.

As an example, we take the quercetin glycosylation pathway in cotyledons, occurring during the development of tomato seedlings (*Koes, Quattrocchio & Mol, 1994*). Quercetin glycosides are a subset of flavonoids, which are plant secondary metabolites naturally produced by plants. Flavonoids are being intensively studied for their proposed beneficial effects on prevention of chronic diseases (*Bovy, Schijlen & Hall, 2007*; *Rein et al., 2006*; *Moon, Wang & Morris, 2006*).

We have measured the concentrations of several quercetin derivative compounds accumulating in cotyledon- and hypocotyl tissues. We have daily measurements from day 5 after sowing up to day 9. The same sample used for the metabolite analysis with liquid chromatography mass spectrometry were used for gene expression analysis. The expression levels of genes, putatively involved in the glycosylation of quercetin, were quantified using microarray analysis. Glycosyltransferases (GTs) are members of the multigene superfamily in plants that can transfer single or multiple sugars to various plant molecules, resulting in the glycosylation of these compounds (*Wang, 2009*). To date, it is not known exactly which GTs catalyze each glycosylation reaction. With more than 200 GT candidates an experimental validation of every single GT is costly. Therefore we wanted to make a pre-selection of the potentially strongest gene candidates, using mathematical modeling and simulations. We use the heuristics that if the kinetic ODE model describes the system of enzymatic reactions reasonably well, the estimated catalytic rates should reflect the real enzymatic activity. This in turn should correlate with the expression trends of the GTs observed using the time series microarray analysis.

Our procedure for the GT inference is as follows:

1. Given the time series metabolite concentration data, estimate the time dependent parameters using all biologically relevant networks. Select the network that gives the best fit to measurements with respect to residual or goodness of fit etc. Save the estimated catalytic rates corresponding to the best network as reference.

2. Compute correlations between the time series of expression levels of each GT and the previously saved series of catalytic rates.

3. Select those GTs whose dynamics correlate best with catalytic dynamics for further experimental validation.

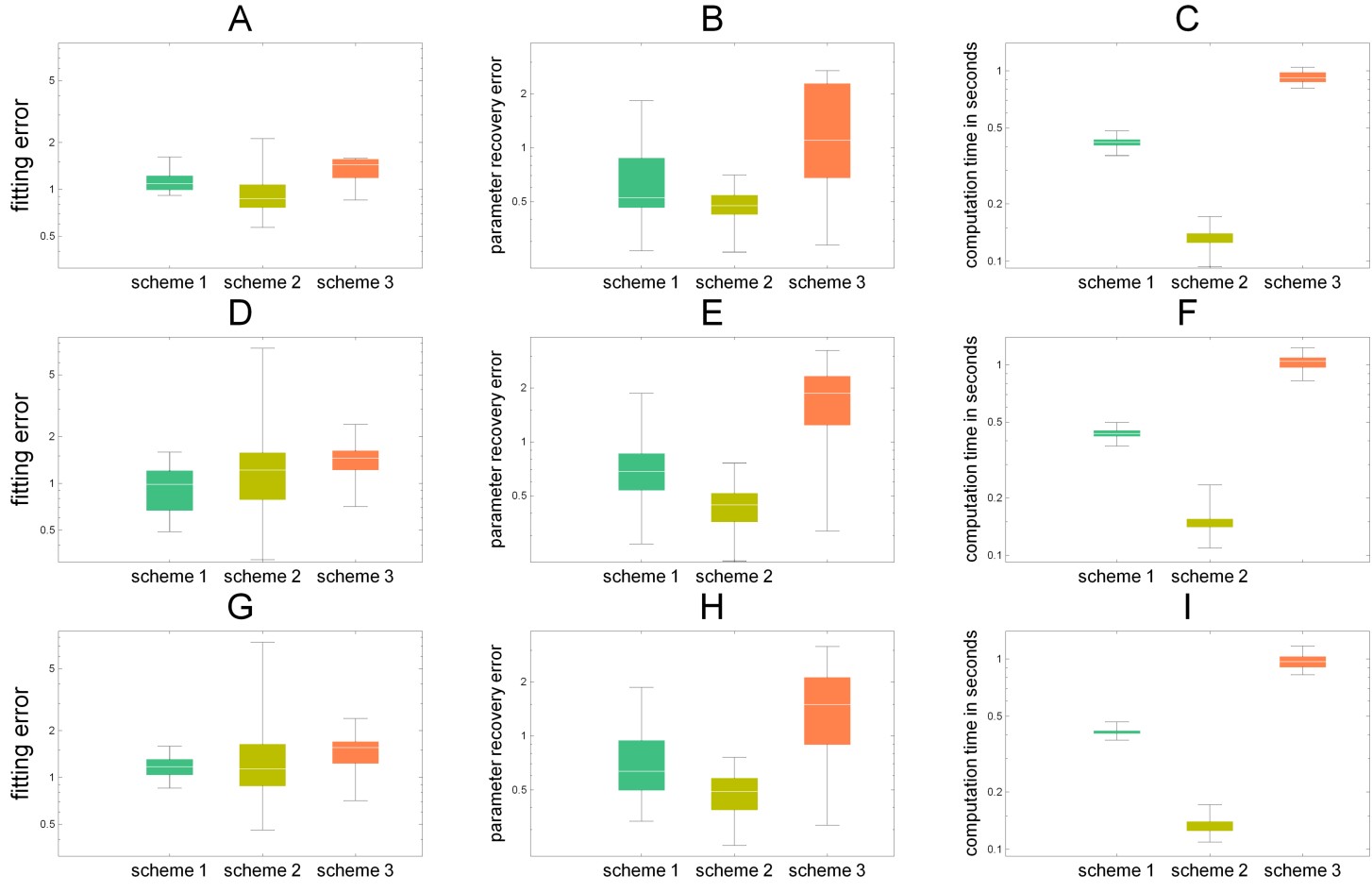

**Figure 4** **We have compared three different reconstruction schemes in 100 simulations, when the underlying network has a tree structure (A–C), with non tree structures (D–F), and with 10% noise added to data (G–I).** In each sub-figure the box plots of simulation results are plotted. (A, D, G) the average point-wise errors in the estimated concentrations. (B, E, H) the average absolute differences in the recovered parameters (catalytic rates) vs. the parameters used to simulate the data. (C) the computation times in seconds. In all figures, the logarithmic scale is used. In terms of network inference, schemes 1 & 2 give in general lower errors.

## RESULTS

### Comparison of parameter inference schemes

As can be seen from Figs. 4A, 4D and 4G, concerning the fitting errors, all schemes give similar results and their box-plots have some overlap. In principle they are solving the same optimization problem, only scheme 1 first solves the point wise rate values and then fits a polynomial, whereas scheme 2 searches for a polynomial-valued rates that fit to the whole series of data and scheme 3 tries simultaneously estimate the parameters as well as the derivatives. We measured the accuracy of the parameter estimation by computing the Frobenius norm (10) of the difference between the original timevariant kinetic rates used in simulation and the reconstructed rates. Besides the actual estimation accuracy, also computation times are relevant. In terms of computation time, scheme 2 is the fastest and scheme 3 is slowest, although the differences are not large. Notice that the comparisons in Fig. 4 were done in a setting where equal parameter constraints ($k_{ij} > 0$) were given

to the solvers and the parameters were estimated using constrained non-linear global optimization (NMinimize in Mathematica) choosing for the fast Nelder–Mead algorithm (with option "PostProcess" → False).

This result is more or less to be expected, since when the data is reasonably accurate, it does not always make sense to re-estimate the data by using it as an unknown variable in the equations of the system. Rather, it may pay off to substitute the data directly into the equations reducing the number of unknown elements. Also it is logical that schemes 1 and 2 perform less well on non-tree graph networks, since the assumption on unique point-wise estimability is not valid anymore. Since our method is based on initial fitting of splines, the major sensitivity is indeed with respect to data. This was also confirmed by the sensitivity analysis we conducted.

Our network models, although relatively small, belong to the general group of the so-called sloppy biochemical models (*Gutenkunst, 2008*), despite of which the parameters still may be identifiable. For a separate discussion and more background on this subject, please see 'Discussion'. The range of eigenvalues of the Hessian of the residual (between predicted and measured values) varies from $10^{-4}$ to $10^5$. For the sensitivity analysis numerical derivatives need to be computed. Since we are considering time varying parameters, we have taken time-averages of point-wise derivatives. Eigenvectors corresponding to very small eigenvalues, implying sloppiness in sensitivity, all point towards those parameters that are associated with network nodes where the measured metabolite concentrations are very low. This is logical since the parameters associated with concentration values close to zero have little effect on the residual, because our objective function does not contain the standard deviation term in the denominator. By this choice we explicitly wanted to avoid that those measurements that are close to noise level shall have equal weight with the more abundant ones.

## Enzyme inference from microarray data

In Fig. 5 we illustrate the results of the analysis as described in 'Parameter inference as a mean to select active genes'. These are the expression levels of best matching six GTs together with the estimated catalytic rates for the reactions that corresponds to the conversions from node $X_i$ to $X_j$ exactly as in Fig. 1A. We have standardized, i.e., subtracted the mean and divided by standard deviation both predicted and measured expressions for visual comparison. As can be seen from Fig. 5, the deviation of the expression levels between samples can vary from gene to gene. One could also weight the correlation according to this variation so that more precise observations are favored. For accurate reconstruction of both the kinetic rates as well as the selection of appropriate genes, a time series with more data points is desired. What exactly the minimal sample number and sampling method should be depends on the data and the system model, but a rule of thumb from experienced modelers would be a minimum of 15 data points. To test experimentally whether the inferred genes are actually related to the enzymes that glycosylate the flavonols, a set of selected genes are currently being cloned.

As a computational validation of the selection procedure, we tested whether substituting the (scaled) expression levels of the selected genes into the model will result in a decreased

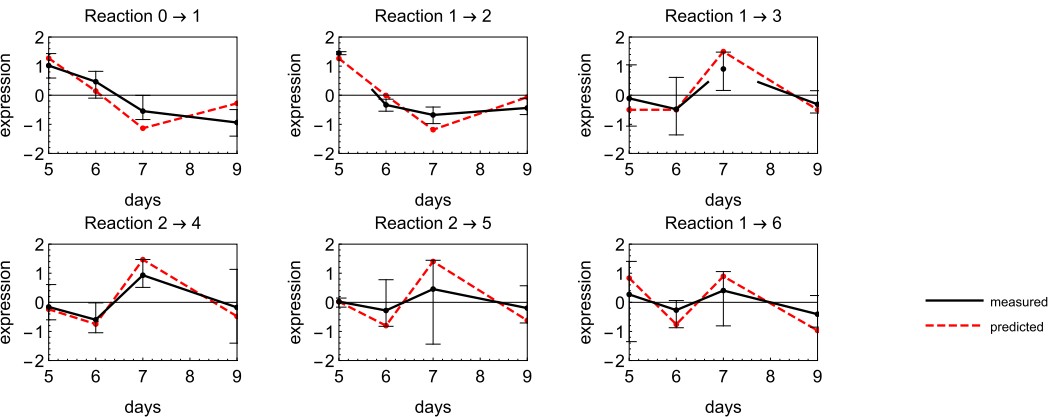

**Figure 5** **The mean expression levels of different glucosyl transferase (GT) candidate genes and the estimated catalytic rates for reactions in a putative network.** Here the best matching gene expression profiles are retrieved from the data.

residual (better likelihood of observing the measurements). The reason we want to do this post-analysis is two-fold. First of all, our GT candidates are ranked according to their correlation with the predicted enzymatic trends, but it may happen that several candidates have almost equal correlation coefficients. This makes it difficult to distinguish between the candidates, especially because the initial GT-population is already a result of an ontology-based selection. Another point is that the selection of the most likely GT's is based on individual matchings with single dynamic parameters whose magnitudes are unknown. It is not absolutely clear, say, whether the combination of the very best candidates will always give better results than when, for example, one candidate is actually the second best one (in terms of correlation). In each network combination, at most seven GTs are considered, but the number of all possible combinations is still very high. Also the expression levels need to be scaled to match the metabolic model.

To ensure a rich set of gene combinations, we ran a Markov Chain Monte Carlo-algorithm (MCMC) (*Calvetti & Somersalo, 2007*). To address the question of whether the differences in correlations are significant enough, we first ordered the genes into a sequence according to their correlation with the predicted enzyme concentration levels and took two sets of genes according to their order number in the sequence: $1, 2, \ldots, 10$ and $11, 12, \ldots, 20$. We tested whether the residuals, obtained after 200 iterations of 1,000 samples with MCMC algorithm using the data of these two sets, have equal means and variances. For the mean test we obtained a $P$-value less than 0.00001 and for the variance test a $P$-value of less than 0.006. We may conclude that in the context of a dynamic kinetic reaction model, those genes with expression levels highly correlating to the predicted enzyme dynamics, are significantly more likely to be responsible for the observations.

## DISCUSSION

In this section we discuss the results in terms of identifiability which is a major issue in parameter inference. A parameter estimation method may always be able to find some
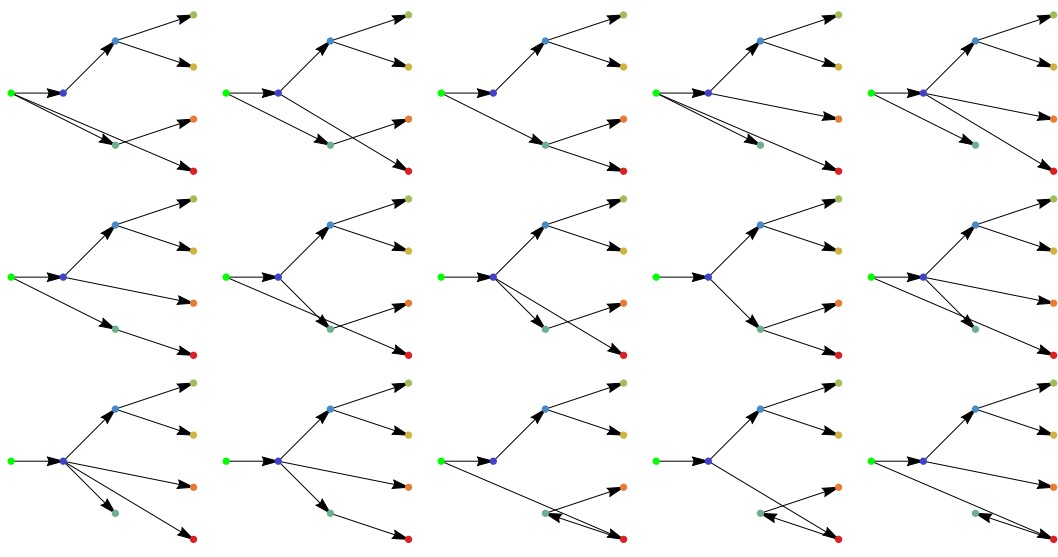

**Figure 6** All biologically feasible networks of the quercetin glycosylation pathway.

estimates, but this makes sense only if it is clear that it is possible to estimate the parameters from the data, i.e., they are structurally and practically identifiable.

## Structural identifiability

A general problem in parameter estimation is that it is difficult and sometimes even impossible to be sure that the estimated parameters are unique. If the model is structurally unidentifiable, there is an infinite number of parameter sets that give equal results. This is a substantial challenge, especially when the network structure is not known, since an overly complex network can result in over-fitting. This problem is not present in any of the (biologically) potential networks as sketched in Fig. 6, since as tree graphs these all turn out to be locally structurally identifiable as they can be embedded in an upper triangular matrix as discussed in the preceding section.

## Practical identifiability

Structural identifiability does not imply practical identifiability and therefore we have studied the practical identifiability of the parameters in our system by means of profile likelihood (*Raue et al., 2009*). We learned that all the kinetic parameters connecting substrates and products with concentrations above detection limit show also practical identifiability (see Supplementary Data). Another observation is that if we allow a product to decay without constraints, the practical identifiability as well as the tree structure of the graph is lost.

## CONCLUSIONS

In this article, we consider the time dependence and unique estimability of kinetic rates in metabolic networks. Firstly, we show that when the underlying network has a structure of a tree graph, these rates can be unambiguously estimated. Secondly we propose a

fast approach for the estimation of time dependent kinetic rates and demonstrate its performance on simulated data. Finally we also propose an application where we utilize the estimation method to detect the genes that are potentially involved in particular enzymatic reactions using microarray data.

### Funding

The Netherlands Consortium for Systems Biology funded this research. The funders had no role in study design, data collection and analysis, decision to publish, or preparation of the manuscript.

### Grant Disclosures

The following grant information was disclosed by the authors:
The Netherlands Consortium for Systems Biology.

### Competing Interests

The authors declare there are no competing interests.

### Author Contributions

- Laura Astola analyzed the data, wrote the paper, prepared figures and/or tables, reviewed drafts of the paper.
- Hans Stigter and Marian Groenenboom analyzed the data, reviewed drafts of the paper.
- Maria Victoria Gomez Roldan performed the experiments, analyzed the data, contributed reagents/materials/analysis tools, reviewed drafts of the paper.
- Fred van Eeuwijk conceived and designed the experiments, reviewed drafts of the paper.
- Robert D. Hall conceived and designed the experiments, contributed reagents/materials/analysis tools, reviewed drafts of the paper.
- Jaap J. Molenaar conceived and designed the experiments, analyzed the data, wrote the paper, reviewed drafts of the paper.

### Data Availability

The raw data has been supplied as a Supplementary Data.

### Supplemental Information

Supplemental information for this article can be found online at http://dx.doi.org/10.7717/peerj.2417#supplemental-information.

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
