# Peer review of "Parameter estimation in tree graph metabolic networks"

_PeerJ, doi:10.7717/peerj.2417_

## Round 0.1 · original submission · Major Revisions

Both reviewers see value in the paper, and I have little doubt about its suitability for PeerJ. I agree with both reviewers, however, that additional description of methods is needed for the article to be suitable for publication. In particular, description of the experimental data used and application of the mathematica notebook provided would help the manuscript. I would also encourage the authors to review the paper for style, as there are a few places where sentences are difficult to read (and probably some places where \citep should be used instead of \cite).

·

Basic reporting

The article is well written and the language is clear.

Comments on figures:

Figures 6/7. The 3d histograms are difficult to read. Some bars are hidden behind other bars. Simple boxplots would convey the same info more clearly. Also, the text descriptions don’t match up with the figure titles. In 6, are the three plots 1) residuals, 2) parameters, 3) computational time, or 1) no noise, 2) noise, 3) computational time. Similar issues in Figure 7. Finally, axis labels would be useful.

Figure 8. Only two graphs are shown, but three are described. This figure is labeled as Figure 7 on line 248. Gene names should be given. Also, it is not clear how samples are ordered on the x-axis. What is time/condition? Are they in order of time or condition? What were the conditions? No conditions are described in section 4.

Experimental design

The research question is well defined and described.

Some of the details of the experimental and analytic methods are not well described:

Sections 2.2 (practical identifiability) and 3.2 (parameter estimation) rely on experimental data. I assume (though it’s not completely clear) that the data used in these sections is the tomato flavonoid data in Section 4. Were these data generated as part of this study? If so, the methods used to generate them should be described in more detail, and preferably provided before sections 2.2 and 3.2. How was the metabolite data generated and pre-processed? The gene expression data? Were multiple conditions used (Figure 8)?

121-125: The results of the practical identifiability analysis should be shown. Where are constraints on product decay given in the model?

Section 3.2:
-The polynomial functions used to model the k_ij(t)’s look like they allow negative values. Were constraints used to prevent these functions from going negative?
- The differences between the three schemes are not clear to me. Is the difference between 3 vs (1 and 2) that in 3 P-splines are not used to calculate derivatives of products?
- In the simulations with different network topologies, was the actual network topology assumed known during the parameter inference step, or was it inferred during the estimation?
- how were estimated parameters compared to the originals in the case of non-tree networks, particularly for schemes 1 and 2 (don’t these assume tree-networks)?

198: Which is “our method”? Scheme 2? All 3? Not clear.
202-204: This sentence is difficult to follow.
267-276. The methods here are not clear. What MCMC algorithm was used? What was the proposal distribution and accept / reject scheme?

Validity of the findings

No comments

·

Basic reporting

In this paper, Laura Astola and collaborators, define a new mathematical modelling method to estimate time varying kinetic rates in metabolic pathways organized in a tree-like structure (i.e. no cycles available in the network topology).
Combination of the method with metabolomics and transcriptomics data could help to detect key genes in metabolic pathways. This is important for example when designing transgenic approaches to introduce metabolic pathways in other species.
The paper is well written, relevant literature has been cited in the introduction and raw data provided, however the whole section on materials and methods needs to be provided. See below.
Figures need to be improved. See comments below.
Even though the paper is quite technical is well explained and should be accessible to a broad audience.

Major comments:

1- There is no Materials and Methods section! A program (DynamicKineticRateSolver) is provided but no information about how to run it is provided. This should be fixed to enable others to replicate the analysis.
2- There are some figures that are provided as individual files but are not included in the main text or commented. Likewise figure 2 in the text is not provided as an individual file.
3- Supplementary files with metabolite information and gene expression is provided and used to construct figures in the text, but is not explicitly mentioned.


Minor Comments:

1- Fig 2, Possible networks of the quercetin glycosilation pathway are presented but no information has been given before in the article about the quercetin pathway. A link to a metabolic database would be helpful. Eg. (http://pmn.plantcyc.org/PLANT/NEW-IMAGE?type=PATHWAY&object=PWY-5321#)
2- Line 118: Please provide a bit more information about why in tree graphs estimated parameters are unique because they can locally structurally identifiable.
3- Fig 3 is very intuitive but guiding the reader in the text through the whole process would be really helpful.
4- Fig 4 Legend. Indicate Fig 1A instead of Figure 1 on the left.Please also refer to Fig A, B, and C, rather than left hand, middle and right hand.
5-Line 145 since we do not have measurements instead of for we have no measurements.
6-Line 147 for which we have instead of on which we have
7-Line 172 Which figure?
8- Figure 6 and 7. Use letters to indicate the different panels. I suggest to use 2D figures for the distribution histograms.
9- Figure 8. Letter size is very small. Right border of left graph is missing. Avoid shading in the legend. Use letters to indicate panels. Is not clear what is represented in the x-axis.
10- line 248. Should be Figure 8 not Figure 7.
11- I suggest to provide supplementary files as individual csv files and provide generated graphs on separate file.

Experimental design

I don´t see any major flaws in the experimental design. As I mentioned above, experimental metabolite and gene expression data provided as supplementary material should be explicitly mentioned in the text.

Validity of the findings

Could the authors comment on the number of data points needed to obtain reliable results and on the time resolution (hours, days)?

Mathematical models well constructed and experimental data seems robust, but methods used to obtain them should be explained.

Additional comments

Sorry for the delay turning out the revision

---

## Round 0.2 · Minor Revisions

Many thanks for your attention to the reviewers' concerns. Reviewer 1 has a few additional comments I believe worth considering. Please consider reviewer 1's thoughts about reorganization and figure 3, but the final decision on whether to do this is up to you. I would ask you to address reviewer 1's remaining question about experimental design. I do not, however, see any need for the paper to go back out to reviewers.

·

Basic reporting

I appreciate the authors' additions in the methods section and the re-made figures.

I would suggest a different organization. I think that sections 1, 2.1, 2.2, 3.1, 5.1, 5.2, plus half of 3.2 and 4 are all methods. Results are given in the 2nd half of 3.2 and 4. I think having the background given in sections 2 and 5 before the results is more useful for a methods paper like this.

Figures 3 is improved. But a couple suggestions: I recommend that each column should have the same y-range so that comparisons among the simulations is easier. Also, since it is interesting to directly compare the three classes of simulations, re-arranging the panels so that the three fitting error panels are in the same row would be useful.

Experimental design

262-283. I’m afraid I still do not understand this section well. I believe the goal is to substitute expression-based estimates of enzyme concentrations for the kinetic rates, and test whether this improves the fit of the model (decreases the residuals). I was expecting an answer like: “when substituting candidate gene expression for kinetic rates, the RMSE decreased by XX%”. Can the analysis or the MCMC algorithm give this kind of a result?

Validity of the findings

No comments

·

Basic reporting

I think the authors have responded adequately to the comments and suggestions made by the reviewers, specially concerning the description of the materials and methods. Raw data and a Mathematica notebook with the data analysis has been provided, although I have not been able to evaluate the notebook since I don´t have a Mathematica license.

Experimental design

No Comments

Validity of the findings

No Comments

Additional comments

No Comments

---

## Round 0.3 · accepted · Accept

Many thanks for the quick turnaround! Changes to figure and order are appreciated, and I'm OK with the explanation for experimental design sections and keeping it as is.